# Qualitative and Quantitative Detection of Food Adulteration Using a Smart E-Nose

**DOI:** 10.3390/s22207789

**Published:** 2022-10-14

**Authors:** Kranthi Kumar Pulluri, Vaegae Naveen Kumar

**Affiliations:** School of Electronics Engineering, Vellore Institute of Technology, Vellore 632014, India

**Keywords:** electronic nose, food, adulteration, artificial neural network, support vector machine, beef, pork, sample slicing window protocol, qualitative, quantitative

## Abstract

Food adulteration is the most serious problem found in the food industry as it harms people’s healths and undermines their beliefs. The present study is focused on designing and developing a smart electronic nose (SE-Nose) for the qualitative and quantitative fast-track detection of food adulteration. The SE-Nose methodology is comprised of a dataset, sample slicing window protocol, normalization, pattern recognition, and output blocks. The dataset pork adulteration in beef is used to validate the SE-Nose methodology. The sample slicing window protocol extracts the early part of the signal. The sample slicing window protocol and pattern recognition models (classification and regression models) together achieved the high-performance and fast-track detection of pork adulteration in beef. With classification models, the qualitative analysis of adulteration is measured, and with regression models, the quantitative analysis of adulteration is measured. An accuracy of 99.996% and an RMSE of 0.02864 were achieved with the SVM classification and regression model. The recognition time in detecting pork adulteration in beef with SVM models is 40 s. With the proposed SE-Nose methodology, the recognition time is reduced by one-third. To validate the classification and regression models, a 10-fold cross-validation method was used.

## 1. Introduction

Growing demand for nutritious, high-quality, and safe food has made the food industry and consumers pay more attention. Food adulteration is the biggest problem in the food supply chain and is prevalent at various levels. Food adulteration, committed wittingly or unwittingly, is a major problem and can have severe consequences for people with food intolerances, lifestyles, and religious practices. The most common reasons for food adulteration entail substituting high-value food products with low-value food products for financial profits [1,2]. Beef is one of the most widely consumed of all meats because it is low in fat, high in protein, vitamins, amino acids, and other essential nutrients required for human beings [3]. According to the United States Department of Agriculture’s (USDA) world market and trade reports, the United States is anticipated to be the world’s top producer and exporter of beef in 2022 [4]. Because of its high consumption and economic benefits, beef is the prime target for adulteration and is most frequently adulterated with pork due to its lower economic value. Consumers object to such adulterations for allergic, cultural, monetary, religious, and consumer rights reasons [5,6]. In an investigation conducted in China on 1553 media reports, of the total instances of adulteration, animal-based food makes up 37.78% [7]. Halal-certified meat pies served to prisoners in UK jail in 2013 were found to contain pork DNA in them [8]. With the horse meat scandal incident in 2013, the demand for beef authentication has increased significantly [9]. With these types of incidents, there is a rise in the need for an accurate and reliable method of detecting beef adulteration.

To detect beef adulteration, various techniques are employed such as molecular biology-based methods [10,11,12,13,14,15,16], spectroscopy methods [3,6,17,18,19,20,21], chromatographic methods [22,23,24], and enzyme-linked immunological methods [25,26,27]. However, the equipment to implement such techniques is expensive, poor in real time, time consuming, destructive, laborious, and requires technical supervision. Additionally, the equipment requires highly skilled individuals to conduct the experiment and analyze the results. This demands the development of an instrument that is simple, accurate, economical, portable, quick, highly sensitive, and requires fewer samples for measurement.

An electronic nose (E-nose) is a device that mimics a human’s sense of smell. It is simple, sensitive, reliable, low-priced, non-destructive, and highly correlated with the detection of various odor signatures. An E-nose is built up of an array of sensors, signal processing units, and a pattern recognition system to measure the testing sample. The sensors that are used for the design of a sensor array are metal-oxide semiconductor (MOS) gas sensors [28], metal-oxide semiconductor field effect transistors (MOSFET) gas sensors [29], acoustic wave gas sensors [30], quartz crystal micro balance sensors [31], infrared sensors [32], colorimetric sensors [33], fluorescence sensors [34], conducting polymer gas sensors [35], and fiber optic gas sensors [36], among others. Data from the sensor array are received by the signal processing unit and pre-processes the data as per the requirement. A pattern recognition system performs the classification or regression of the processed data to measure the sample. The development at all stages of E-nose design made it a good alternative to traditional methods. E-nose is employed in different food industries such as the meat industry [37,38], oil industry [39], alcohol industry [40], tea industry, [41] etc. Furthermore, the E-nose is used in various food industry applications such as adulteration detection [42], origin identification [43], authentication [44,45], grading [46,47], freshness assessment [40], and shelf-life determination [48]. In this study, we develop a smart E-Nose (SE-Nose) for the qualitative and quantitative analysis of pork adulteration in beef. The smartness of the E-Nose is achieved by introducing a sample slicing window protocol (SSWP) for fast-track detection, designing a less complex system, qualitative analysis of adulteration using artificial intelligence-based classification models, and the quantitative analysis of adulteration using regression models, and achieving higher accuracies. In other parts of this section, an extensive literature analysis of evaluating the E-nose to identify beef adulterated with pork is described.

E-nose, composed of ten semi-conductive polymer sensors with multi-discriminant analysis, is used to identify beef–pork mixtures in four compositions [49]. The cooked and uncooked samples are stored at 4 °C for 0–6 days. The sensing time for measuring each sample is of 4 min. The presented results clearly show that the designed E-nose identifies samples according to mixture composition and storage days. FOX4000 E-nose with discriminant factor analysis (DFA), principal component analysis (PCA), and partial least square analysis (PLS) techniques are used to detect pork adulteration in beef [50]. The results demonstrate that the system is capable of detecting pork adulteration in beef in various ratios. The coefficient of determination (R2) between sensor data and minced pork ratios is 0.9762. To study and find the optimal temperature required for the measurement sample to be tested using E-nose for discriminating beef–pork mixtures, a sensory array made of eight MOS sensors with temperature and humidity sensors is designed [45,51]. Five beef–pork mixture classes are discriminated in this study using a support vector machine (SVM), k-nearest neighbors (KNN), random forest, and naive Bayes techniques at −22 °C, room temperature, and 55 °C, respectively. According to this study, continuous gases from beef and pork can be detected at −22 °C with maximum accuracy to detect beef adulterated with pork. Using the same sensor array, another author identified seven beef–pork mixtures with an optimized E-nose system designed using a sensor array followed by noise filtering and optimized SVM blocks [42]. Each mixture sample is measured for 15 min, and 60 data points are recorded for each measurement. An accuracy of 98.10% is obtained with the designed system. To analyze the effect of gas concentration in the detection and classification of beef–pork mixture, an E-nose consisting of nine MOS sensors is used [52]. The proportion of beef–pork mixture used is the same as that used by [42]. Sample sizes of 50 mL, 150 mL, and 250 mL is used for measurement, and each sample is measured for 2 min. Statistical feature extraction, classification, and ensemble learning are implemented on the measured data to determine the system’s performance. Good classification values are obtained using the ensemble learning method with 50 mL. An E-nose designed with an array of colorimetric sensors with twelve chemical dyes was developed to detect beef adulterated with pork [33]. The twelve chemical dyes of the colorimetric sensor array are selected with chemically receptive dyes that can detect complex volatile organic compounds (VOCs) generated from meat samples. The colorimetric sensor array is placed in the sample chamber for five minutes so that the dyes on the sensor array react with the VOC from the meat sample. The classification methods Fisher linear discriminant analysis (Fisher LDA) and extreme learning machine (ELM) are used to classify the three mixture combinations of pork adulteration in beef. The ELM method yields an accuracy of 87.5% with the prediction set. Additionally, the adulteration level is calculated using a backpropagation artificial neural network (BP-ANN), and a correlation coefficient (R) and root mean square error (RMSE) of 0.85 and 0.147 are obtained, respectively.

It is clear from the literature discussed above that the E-nose can predict beef adulterated with pork. The problems identified in the literature are the longer measurement time, low classification accuracy, lesser beef–pork mixture ratios, the complex design of architecture, and the missing quantitative analysis. This study developed the SE-Nose methodology for the qualitative and quantitative fast-track detection of pork adulteration in beef. The implementation of SE-Nose is performed in four stages: data collection, pre-processing, pattern recognition, and output. The data collection stage deals with the dataset used to validate the methodology, and the pork adulterated with beef dataset available in the literature [53,54] was used in our study. This dataset helps to overcome the problem of a less beef–pork mixture ratio identified in the literature, as this dataset contains seven combinations of beef–pork mixture ratios. The pre-processing stage contains SSWP and normalization blocks. The SSWP extracts the early part of the signal to achieve fast-track detection, thereby overcoming the problem of a longer measurement time and complex architecture problems identified in the literature. In the pattern recognition stage, multiple classifications and regression models are implemented for the classification and identification of adulteration levels present in the sample. With the design parameters of pattern recognition models (SVM, ANN), the problems of low-classification accuracy and complex architectures are solved. The output stage publishes the qualitative and quantitative analysis of sample adulteration using the results obtained from the pattern recognition model. The performance of the SE-Nose system is evaluated using the size of input data, the count of windows, recognition time, training time, validation time, accuracy, mean square error (MSE), RMSE, R, and R2. Furthermore, the results obtained with the SE-Nose system are compared with the results present in the available literature.

The organization of the remaining sections of the paper is as follows: Section 2 discusses the SE-Nose methodology consisting of pork adulteration in the beef dataset, SSWP, normalization, classification models, regression models, and outputs. Section 3 discusses the results obtained with SSWP, classification models, and regression models for the qualitative and quantitative analysis of pork adulteration in beef. Furthermore, Section 3 compares the results obtained with the SE-Nose methodology with comparable results available in the literature. Section 4 presents the conclusion of our study.

## 2. Methodology

The block diagram of the proposed SE-NOSE is shown in Figure 1. The working principle of SE-Nose is divided into four stages, namely data collection, pre-processing, pattern recognition, and output. The dataset used in this work is made available in the data collection stage. To test and validate the SE-Nose, pork adulteration in the beef dataset is used. The next stage is the pre-processing stage, and here the data are processed using SSWP and normalization. The next stage is the pattern recognition stage. In this stage, several classification and regression models are used to classify and predict the level of adulteration. The last stage of SE-Nose is the output stage. This includes the qualitative and quantitative analysis of the results obtained from the pattern recognition stage. The other part of this section explains the seven blocks of the four-stage SE-Nose.

### 2.1. Pork Adulteration in Beef Dataset

The dataset generated by R. Sarno et al. [53,54] for evaluating pork adulteration in beef is used to implement the proposed methodology. The dataset is available at the URL: “https://data.mendeley.com/datasets/5yhggs7zy7/1, accessed on 1 July 2022”. The adulteration is performed in various mixing ratios. For each measurement, the sample quantity is 100 g. The beef sample is mixed with pork in seven combination ratios as 0%, 10%, 25%, 50%, 75%, 90%, and 100%. The sensor array for detecting VOC from the samples contains sensors, namely MQ138, MQ137, MQ136, MQ135, MQ9, MQ6, MQ4, MQ2, and DHT22. The temperature and humidity is measured using DHT22 sensor. The selectivity of each sensor is detailed in Table 1. At a sampling frequency of 0.5 Hz, each sample is measured for 120 s, yielding 60 records for each measurement and 60 measurements for each sample ratio. Thus, for seven beef–pork mixture ratios, 420 measurements were conducted and made available as a dataset.

### 2.2. Sample Slicing Window Protocol (SSWP)

The SSWP proposed in this paper extracts the early part of the signal and provides it as output. Here, the early part of the signal means the initial portion of the signal. The early part of the signal is framed as the window. In Figure 2, we can observe the whole set of signals represented in Figure 2a, and the corresponding early part of the signals is framed as windows in Figure 2b and Figure 2c, respectively. With the SSWP, the fast-track detection of food adulteration is possible as pattern recognition models are provided with the early part of the signal over the complete signal in traditional methods, which take a long time for detection. The size of the early part of the signal depends on the window number and step size. The mathematical representation of the SSWP is discussed below:

The SSWP’s input is sensor data from the dataset. It is represented as x[i,j], and the output is represented as y[i,j]. Here, *i* represents the count of sensors present in the sensor array, and *j* represents the measurement duration. Equation (Equation 1) represents the early part of signal extraction.
(1)y[i,j]=x[i,0:(w×Δs)]

In this equation, Δs indicates the signal’s step size and is a smaller part of the signal. After numerous experimental simulations, the value of Δs is chosen as 10 for detecting pork adulteration in beef using the dataset discussed in Section 2.1. ‘*w*’ indicates the windows count for adulteration detection and is calculated using Equation (Equation 2).
(2)w=1,2,3,......,k

Here, ‘*k*’ stands for the total count of windows employed and is determined using Equation (Equation 3).
(3)$$k=\frac{n}{\Delta s}$$

In this Equation, ‘*n*’ stands for the count of records in a measurement. Here, ‘*n*’ is 60 for the chosen dataset. By substituting the parameter values in Equation (Equation 3), the window length ‘*k*’ is determined to be 6.

The working mechanism of the SSWP is presented using case 1 and case 2, representing the first and last windows, respectively, i.e., the first and sixth windows for pork adulteration in beef, using the parameters defined above.

Case 1: We discuss first window implementation in case 1. To quickly detect the pork adulteration in beef, the first window data (first slice of the data) are extracted from the entire dataset. Equation (Equation 4) represents the data of the first window obtained by substituting required parameters in Equation (Equation 1).
(4)y[i,j]=x[i,0:(1×10)]=x[i,0:10]

Case 2: Here, in case 2, we implement the last window, i.e., the sixth window. As this is the last window, the entire measurement data are considered for sixth window. Equation (Equation 5) represents the data of the sixth window obtained by substituting the required parameters in Equation (Equation 1).
(5)y[i,j]=x[i,0:(6×10)]=x[i,0:60]

The extraction of the signal’s early part using SSWP for the first and sixth window is shown in cases 1 and 2. Similarly, the extraction of the early part of the signal for other corresponding windows can be performed. Figure 2 illustrates the extraction of the early part of the signal using sample SSWP for windows 1 and 6, where the coloured lines represents different sensor signals.

### 2.3. Normalization

On the data extracted from the SSWP, data normalization is performed. Using the normalization, the data are scaled in the 0–1 range without affecting the difference in the range of values and loss of important information, thereby preserving data homogeneity. The mathematical expression to implement normalization is:(6)$$z\left[i,j\right]=\frac{y[i,j]-min(y[i])}{max(y[i])-min(y[i])}$$

### 2.4. Classification Models

Classification models SVM-C and ANN-C are implemented on the normalized data generated from the pre-processing stage for the qualitative analysis of adulteration, representing whether the samples are pure or adulterated.

#### 2.4.1. SVM-C

SVM-C indicates that the SVM model can solve the classification problem. SVM is a supervised classification model that maps the input data points to high-dimensional space using kernel functions such as a radial basis function (RBF), Gaussian, second-order polynomial, and linear. The data points are then divided into several classes using a hyperplane to determine which class has the largest margin size. The data point closest to the hyperplane is called the support vector [55]. The two key parameters that influence the performance of the SVM algorithm are gamma (γ) and regularization parameters (*C*). The regularization parameter trade-off between the accurate classification of the training set against maximizing decision margins. The Gamma parameter determines the impact of a single training set. The determined kernel function for our work is RBF with γ = 1 and *C* = 5. The data division for training and testing the SVM-C models is 70% and 30%, respectively.

#### 2.4.2. ANN-C

ANN-C signifies the ANN model to solve the classification problem. It is a supervised classification model that processes data layer by layer, with each layer made up of a large number of neurons, and each neuron modifies data using an activation function, bias, and weights connected to it [56]. A multi-layer perceptron (MLP) with three layers, namely an input layer, hidden layer, and output layer, is used. The input layer contains 10 neurons corresponding to 10 sensor inputs, the output layer contains 3 neurons corresponding to 3 classes of adulteration levels, and finally, based on experimental the simulation results, the hidden layer is designed with 15 neurons. Data divisions of 70%, 10%, and 20% are used for the training, validation, and testing of the ANN-C models.

### 2.5. Regression Models

Regression models SVM-R and ANN-R are implemented on normalized data generated from the pre-processing stage for the quantitative analysis of the level of adulteration, representing how much adulteration has happened.

#### 2.5.1. SVM-R

SVM-R indicates the SVM model to solve the regression problem. SVM-R uses the same principle as SVM to predict discrete values. It finds the best fit line and is the hyperplane with a maximum number of points. The RBF kernel function with γ = 0.5, and *C* = 5 is used in this work. The data division for training and testing the SVM-R models is 70% and 30%, respectively.

#### 2.5.2. ANN-R

ANN-R signifies the ANN model for solving regression problems. ANN-R uses the same principle and generates a function with input variables to predict the output. MLP with three layers, namely the input layer (10 neurons), hidden layer (15 neurons), and output layer (3 neurons), is used. Data divisions of 70%, 10%, and 20% are used for the training, validation, and testing of the ANN-R models.

### 2.6. Output

The last stage of the proposed SE-Nose is the output stage. Two different analytical results are obtained using classification and regression models. Qualitative analysis is obtained using classification models, and quantitative analysis is obtained using regression models.

#### 2.6.1. Qualitative Analysis

The classification models are used for the qualitative analysis of pork adulteration in beef. In the qualitative analysis, we predict or classify whether the beef sample is adulterated or not using three different classes labeled Class-A, Class-B, and Class-C. The description of each class is detailed in Table 2.

The seven classes mentioned in the dataset in Section 2.1 are regrouped into 3 classes, as discussed in Table 2. Here, group labeled class 1 in the actual dataset is now labeled class-A because this group contains the pure beef sample (100% beef). The groups labeled classes 2–6 of actual datasets are now labeled as class-B, as all the data under this label come under the adulteration group. The class 7 labeled group of the actual dataset is labeled as class-C, as this group contains pure pork samples (100% pork).

#### 2.6.2. Quantitative Analysis

The regression models are used for the quantitative analysis of pork adulteration in beef. In quantitative analysis, we predict the amount of pork adulteration in the beef present in the sample in the range of 0–1, representing 0%–100%. The seven mixture combinations of pork adulteration in beef mentioned in the dataset in Section 2.1 are 0%, 10%, 25%, 50%, 75%, 90% and 100% are converted into 0, 0.1, 0.25, 0.5, 0.75, 0.9, and 1 for performing the regression to obtain quantitative analysis. This helps us predict adulteration levels more efficiently with regression.

## 3. Results

The results section discusses the implementation of the SE-Nose methodology on pork adulteration in the beef dataset. Initially, the implementation of the SSWP is presented, followed by the performance of classification and regression models emphasizing the performance achieved at the best and last window. Additionally, the efficiency of the SE-Nose methodology is analyzed with comparable works available in the literature.

### 3.1. Sample Slicing Window Protocol (SSWP)

To the pork adulteration in beef dataset discussed in Section 2.1, SSWP detailed in Section 2.2 is implemented. The whole dataset contains 420 measurements, with each measurement having 60 records. After multiple experimental simulations on step size, Δs is assigned with a value of 10, generating six windows. Using Equation (Equation 1), the early part of the signals for the concerned window is generated, and the early part of the signals for all six windows required in this study is represented in Equation (Equation 7).
(7)$$\begin{array}{cc} y[i,j] & =x[i,0:(1\times 10)]=x[i,0:10]\\ \; & =x[i,0:(2\times 10)]=x[i,0:20]\\ \; & =x[i,0:(3\times 10)]=x[i,0:30]\\ \; & =x[i,0:(4\times 10)]=x[i,0:40]\\ \; & =x[i,0:(5\times 10)]=x[i,0:50]\\ \; & =x[i,0:(6\times 10)]=x[i,0:60] \end{array}$$

### 3.2. Classification Models

To perform the qualitative analysis by detecting whether beef is adulterated with pork, two classification models, SVM-C and ANN-C, were employed in our study. The input to the classification models is the normalized early portion of data generated from the SSWP.

Using the SVM-C model, a maximum accuracy of 99.996% is obtained at the second window, and an accuracy of 99.994% is obtained with the entire data, i.e., with the last window data as input. The recognition time to obtain the best results at the second and last, i.e., sixth window, is 40 s and 120 s. The same can be depicted in Table 3. The maximum accuracy is achieved with the early portion of data recorded at an initial 40 s (second window), thereby reducing the recognition time from 120 s to 40 s.

Using the next classification model ANN-C, a high accuracy of 99.996% was obtained at the third window with a recognition time of 60 s, and an accuracy of 99.993% is obtained at the last window, i.e., the sixth window with a recognition time of 120 s. Table 4 shows the accuracy, window number, and recognition time obtained at the best and last windows using the ANN-C classification model. From the table, it is clear that the highest accuracy is achieved with the early portion of data recorded at the initial 60 s (third window), thereby reducing the recognition time from 120 s to 60 s.

A comparison of the performance metrics achieved using classification models ANN-C and SVM-C is shown in Table 5. As shown in the table, although the predicted accuracy is the same with both ANN-C and SVM-C models, with the recognition time and window number metrics, it is clear that the SVM-C model detected the adulteration in beef earlier than the ANN-C model. With the SVM-C model, the highest performance is achieved at window number 2, and with the ANN-C model, the highest performance is achieved at window number 3. Here, window number 2 means that the data were acquired from the sensor in the initial 40 s of the measurement and window number 3 means that the data were acquired from the sensor at the initial 60 s of the measurement. With this, it is clear that the SVM-C model has the lowest recognition time and the same is visible in Figure 3. The ANN-C model took 20 s longer to recognize than the SVM-C model, and the same can be observed in the red line in Figure 3a. Similar trends are also observed concerning the window numbers and input data size. The window number and size of the input data in the bar and line plots are presented in Figure 3b in blue and black colors. The time required to train the classification models SVM-C and ANN-C are 1.57 s and 4.89 s, respectively, as presented in Figure 3c.

### 3.3. Regression Models

The performances of two regression models, SVM-R and ANN-R, were employed for the quantitative analysis of pork adulteration in beef, representing the amount of pork that is mixed into the beef sample. The input to the regression models is a normalized early portion of data generated from the SSWP.

With the implementation of the SVM-R model, a lower MSE value of 0.00082, and a higher R2 value of 0.993 is acquired at second window, whereas at the last window (sixth window), an MSE value of 0.000929 and R2 value of 0.991 are acquired. The last window represents the whole data. The recognition time to acquire output at second window is 40 s, and at sixth window is 120 s. Table 6, shown below, presents the performance metrics (MSE, RMSE, R, R2) obtained for the best and last windows, their corresponding window numbers, and their recognition times using the SVM-R regression model. Better performance was observed with the early portion of data recorded at an initial 40 s, thereby reducing the recognition time from 120 s to 40 s.

The next regression model used in our study is the ANN-R model. Using the ANN-R model, at the third window, the low MSE of 0.001142 and the high R2 value of 0.987 is achieved with a recognition time of 60 s and at the last window (sixth window) MSE of 0.00142, the R2 value of 0.984 is achieved with a recognition time of 120 s. The last window represents the complete data recorded in a measurement. The details of the performance metrics (MSE, RMSE, R, and R2), window numbers, and recognition times at the best and last window using the ANN-R model are shown in Table 7. From Table 7, it is clear that a high accuracy is achieved with the early portion of data recorded at the initial 60 s (third window) by reducing the recognition time from 120 s to 60 s.

Table 8 presents the comparison of performance metrics obtained using regression models SVM-R and ANN-R for the effective quantitative analysis of the level of pork adulteration in beef. The table shows that lesser MSE, RMSE values, and higher R and R2 values are achieved with the SVM-R model over the ANN-R model. The same is visible in Figure 4a,b. With the SVM-R and ANN-R models, the input data sizes of 20 records and 30 records are observed, and the time required to train the two models is 0.51 s and 3.95 s, respectively. The recognition time and size of the input data in the bar and line plots are presented in Figure 4c. With the comparison of performances between SVM-R and ANN-R, it is clear that the SVM-R model surpasses the ANN-R model in terms of the quantitative analysis predicting the level of pork adulteration in beef.

### 3.4. Comparison and Discussion

In this subsection, the proposed SE-Nose methodology is compared with three similar works available in the literature, and a discussion of our study is presented. The first comparison is performed with R. Sarno et al. [42]. The basis of the comparison is that the dataset used by R. Sarno et al. [42] and our study dataset were similar. The second comparison was performed with Peng et al. [57]. The main reason for this comparison is that Peng et al. [57] used raw data for pattern recognition, similarly to in our study. For the last comparison, the results of F. Han et al. [33] were used in this study, as both F. Han et al. [33] and our study is focused on the detection of pork adulteration in beef.

The results obtained using the proposed SE-Nose methodology were compared with those obtained by R. Sarno et al. [42] and are shown in Table 9. On the data obtained from sensors, the author implemented discrete wavelet transform for noise filtering, statistical feature extraction, PCA for dimensionality reduction, and SVM for classification. A total of seven classes are used for classifying adulteration. An accuracy of 98.10% is obtained to identify pork adulteration in beef with a recognition time of 120 s. The graphical representation of accuracy, recognition time, and input data size is shown in Figure 5a, b, and c, respectively. Finally, from Table 9 and Figure 5, it is clear that, over the results shown by R. Sarno et al. [42], the proposed SE-Nose methodology, there is a rise in accuracy, a decrease in the number of inputs, and a significant decrease in the recognition time, thereby achieving the fast-track detection of pork adulteration in beef with higher accuracies.

Another comparison is tabulated in Table 10 between the results obtained using the SE-Nose methodology and the results reported by Peng et al. [57]. The basis of this comparison is that both the SE-Nose methodology and Peng et al. [57] consider raw sensor data for pattern recognition. The major advantage of this study is that the author applied a deep convolutional neural network for gas classification on the raw sensor data obtained from the sensor array. The main drawback of this work is that the entire signal is used to implement gas classifications, which increases the recognition time of measurement. Additionally, it is evident that, with the SE-Nose methodology, an accuracy of 99.996% is achieved compared to 95.20% with Peng et al. [57]. Figure 6a–c shows the graphical representation of the comparison of performance metrics such as accuracy, recognition time, and training time. With this comparison, it is evident that the SE-Nose method performs fast and accurately.

Two studies focused on detecting pork adulteration in beef are compared and tabulated in Table 11. This comparison shows the results of the proposed SE-Nose method with the results of F. Han et al. [33] that studied the detection of pork adulteration in beef. The PCA method is applied in this study to the features extracted from the colorimetric sensor array, followed by Fisher LDA, ELM methods for classification analysis, and BP-ANN for regression analysis. The main disadvantage of this study is that, for each measurement, the dyes on the colorimetric sensor array are exposed for five minutes to a food sample to detect VOCs from the food sample. Figure 7a–c show the graphical representation of performance metrics accuracy, R, RMSE, and recognition time. From Table 11 and Figure 7 it is clear that higher accuracy, R values, and lower RMSE values are obtained with the SE-Nose methodology compared to in F. Han et al. [33].

With the above study, it is clear that pork adulteration in beef can be detected using an E-nose. The problems identified, such as the longer measurement time, low classification accuracy, lesser beef–pork mixture ratios, complex architecture design, and missing quantitative analysis, were addressed using SE-Nose methodology for the qualitative and quantitative analysis of pork adulteration in beef. The SE-Nose methodology improves performance by reducing the recognition time while minimizing the aforementioned problems. For a qualitative analysis of pork adulteration in beef using the SVM-C classification model, an accuracy of 99.996% was obtained with a recognition time of 40 s. For the quantitative analysis of pork adulteration in beef using the SVM-R regression model, an MSE of 0.00082, RMSE of 0.02864, R of 0.977, and R2 of 0.993 were obtained with a recognition time of 40 s. Furthermore, the SE-Nose methodology can be applied in other food industries.

## 4. Conclusions

The proposed SE-Nose methodology in this study is fast, robust, and effective for detecting and identifying pork adulteration in beef. The qualitative and quantitative analysis of pork adulteration in beef is discussed in this paper. The SSWP proposed in this paper effectively decreases the recognition time of measurement. From the implemented classification and regression models, SVM-C and SVM-R achieve a stronger performance with an accuracy of 99.996%, MSE of 0.00082, RMSE of 0.02864, R of 0.997, and R2 of 0.993. The observed time to train the SVM-C and SVM-R models are 1.57 s and 0.51 s. The recognition time of pork adulteration in beef with the proposed models is 40 s. The classification and regression models are validated using a 10-fold cross-validation method. The results obtained with the SE-Nose methodology were evaluated using the same dataset work and other comparable works available in the literature. Furthermore, the proposed SE-Nose methodology can be extended to other food industry products.

## Figures and Tables

**Figure 1 sensors-22-07789-f001:**
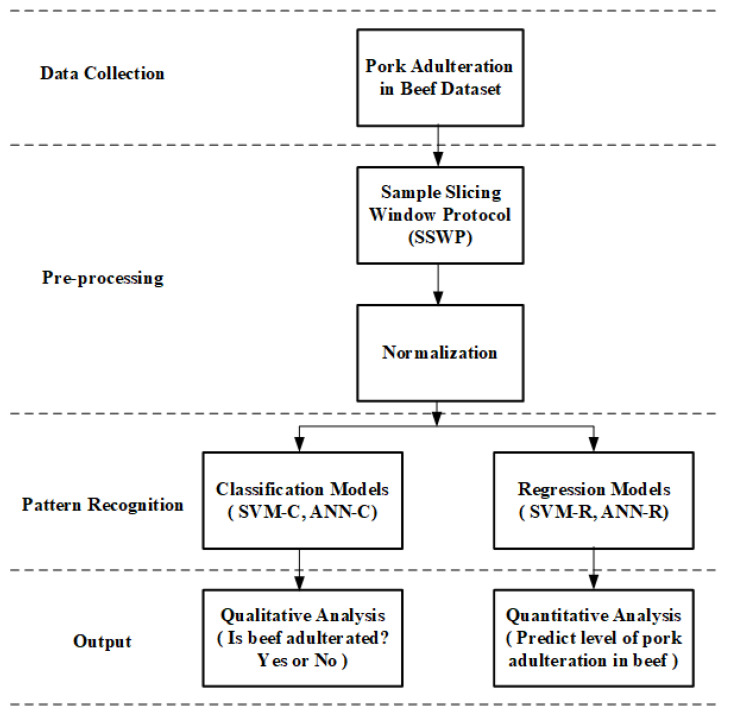
Block diagram of the proposed SE-Nose methodology.

**Figure 2 sensors-22-07789-f002:**
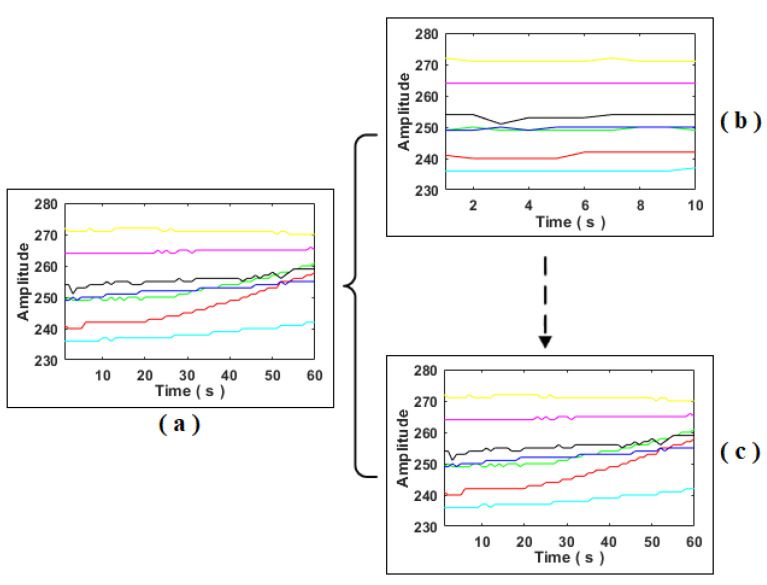
Sample slicing window protocol (**a**) whole data; (**b**) first window; and (**c**) sixth window.

**Figure 3 sensors-22-07789-f003:**
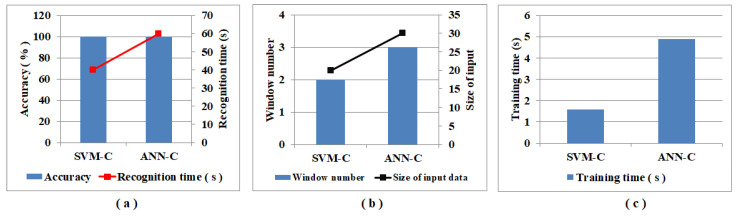
Graphical representation of a comparison of performance metrics (**a**) accuracy, recognition time, (**b**) window number, size of input data, and (**c**) training time of classification models SVM-C and ANN-C.

**Figure 4 sensors-22-07789-f004:**
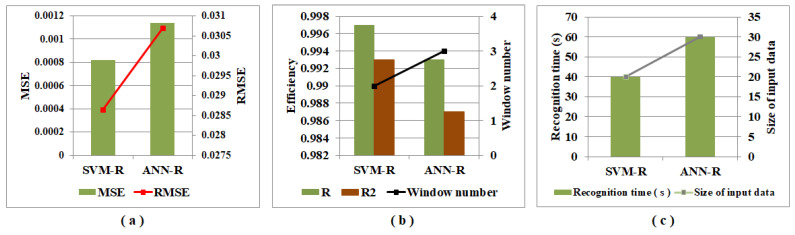
Graphical representation of a comparison of performance metrics (**a**) MSE, RMSE (**b**) R, R2, window number (**c**) recognition time, size of input data of regression models SVM-R, and ANN-R.

**Figure 5 sensors-22-07789-f005:**
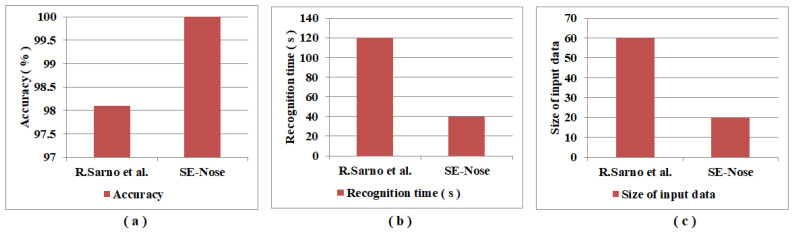
Graphical representation of a comparison of performance metrics: (**a**) accuracy; (**b**) recognition time; (**c**) size of the input data of R. Sarno et al. [42] and the SE-Nose methodology.

**Figure 6 sensors-22-07789-f006:**
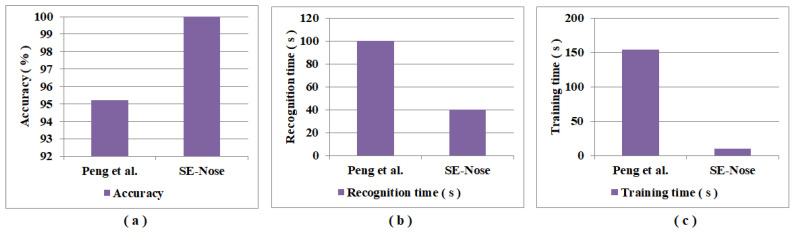
Graphical representation of a comparison of performance metrics: (**a**) accuracy; (**b**) recognition time; and (**c**) the training time of Peng et al. [57] and the SE-Nose methodology.

**Figure 7 sensors-22-07789-f007:**
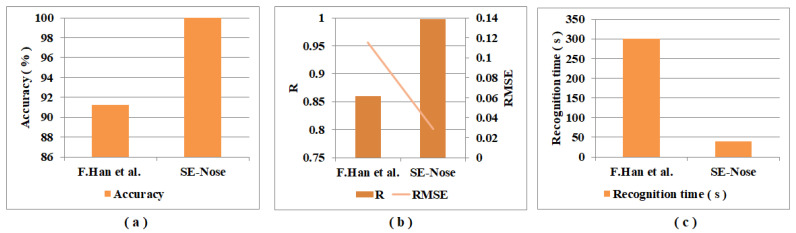
Graphical representation of the comparison of performance metrics: (**a**) accuracy; (**b**) R, RMSE; and (**c**) recognition time of F. Han et al. [33] and the SE-Nose methodology.

**Table 1 sensors-22-07789-t001:** Detailed description of the selectivity sensors in the sensor array.

Sensor	Selectivity
MQ 138	Toluene, acetone, ethanol
MQ 137	Ammonia (NH${}_{3}$)
MQ 136	Hydrogen sulfide (H${}_{2}$S)
MQ 135	NH${}_{3}$, nitrogen, alcohol, benzene, smoke, CO${}_{2}$
MQ 9	Methane (CH${}_{4}$), propane, CO
MQ 6	LPG, iso-butane, propane
MQ 4	CH${}_{4}$, natural gas
MQ 2	LPG, CH${}_{4}$, propane, i-butane, alcohol, hydrogen, smoke
DHT 22	Temperature and humidity

**Table 2 sensors-22-07789-t002:** Detailed description of classes labeled Class-A, Class-B, and Class-C.

Label	Description
Class-A	Pure beef
Class-B	Beef adulterated with pork
Class-C	Pure pork

**Table 3 sensors-22-07789-t003:** Summary of performance of SVM-C algorithm.

Parameter	Best Window	Last Window
Accuracy	99.996	99.994
Window No.	2	6
Recognition time (s)	40	120

**Table 4 sensors-22-07789-t004:** Summary of the performance of ANN-C algorithm.

Parameter	Best Window	Last Window
Accuracy	99.996	99.993
Window No.	3	6
Recognition time (s)	60	120

**Table 5 sensors-22-07789-t005:** Performance comparison of classification models.

Parameter	SVM-C	ANN-C
Accuracy (%)	99.996	99.996
Window No.	2	3
Recognition time (s)	40	60
Training time (s)	1.57	4.89
Validation time (s)	Less than 1	Less than 1
Size of input data	20	30

**Table 6 sensors-22-07789-t006:** Summary of performance of SVM-R algorithm.

Parameter	Best Window	Last Window
MSE	0.000820	0.000929
RMSE	0.02864	0.030479
R	0.997	0.995
R2	0.993	0.991
Window No.	2	6
Recognition time (s)	40	120

**Table 7 sensors-22-07789-t007:** Summary of performance of ANN-R algorithm.

Parameter	Best Window	Last Window
MSE	0.001142	0.00142
RMSE	0.030698	0.036661
R	0.993	0.991
R2	0.987	0.984
Window No.	3	6
Recognition time (s)	60	120

**Table 8 sensors-22-07789-t008:** Performance comparison of regression models.

Parameter	SVM-R	ANN-R
MSE	0.00082	0.001142
RMSE	0.02864	0.030698
R	0.997	0.993
R2	0.993	0.987
Window No.	2	3
Recognition time (s)	40	60
Training time (s)	0.51	3.95
Validation time (s)	Less than 1	Less than 1
Size of input data	20	30

**Table 9 sensors-22-07789-t009:** Comparison of the results obtained by SE-Nose methodology with R. Sarno et al. [42].

Parameter	R. Sarno et al. [42]	Proposed Work (SE-Nose)
Accuracy (%)	98.10	99.996
Recognition time (s)	120	40
Pre-processing	NF+FE+DR	NR
Size of input data	60	20
Classification algorithm	SVM	SVM-C
Learning method	Supervised	Supervised
Number of classes	7	3

Note: NF: noise filtering; FE: feature extraction; NR: normalization; and DR: dimensionality reduction.

**Table 10 sensors-22-07789-t010:** Comparison of results obtained by SE-Nose methodology with Peng et al. [57].

Parameter	Peng et al. [57]	Proposed Work (SE-Nose)
Accuracy (%)	95.20	99.996
Recognition time (s)	100	40
Sensor technology	MOS	MOS
Training time (s)	154	–
Learning method	Supervised	Supervised
Classification algorithm	DCNN	SVM-C
Application	Gas classification	Pork adulteration in beef

**Table 11 sensors-22-07789-t011:** Comparison between the results obtained by SE-Nose methodology and by F. Han et al. [33].

Parameter	F. Han et al. [33]	Proposed Work (SE-Nose)
Accuracy (%)	91.27	99.996
R	0.86	0.997
RMSE	0.115	0.02864
Recognition time (s)	300	40
Sensor technology	Colorimetric	MOS
Learning method	Supervised	Supervised
Classification algorithm	ELM	SVM-C
Regression algorithm	BP-ANN	SVM-R

## Data Availability

https://data.mendeley.com/datasets/5yhggs7zy7/1 (accessed on 1 July 2022).

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
