# Peer review of "Qualitative and Quantitative Detection of Food Adulteration Using a Smart E-Nose"

_sensors, 2022, doi:10.3390/s22207789_

Round 1

Reviewer 1 Report

Tittle :  Why  did you claim that this smart E- nose to detect food adulteration ? Because your sample only beef pork

In which part did you explain the definition of smart E-nose ??

it will be better if the prototype of E-nose can be presented

Line 13 : it will be better if the implication of this study can be included at the end of abstract.

Table 1 : Where Class-C ?? please be careful

Line 125 : What kind of sensors did you use in E- Nose ?

Line 135 : How the condition of sensor and sample (beef and pork) , when pork adulteration was measured ? explain comprehesively  

Line 201 : What software did you use to generate ANN-C, SVM-R ?

Table 7 : table 5 and 6 should be deleted since the information has been presented in table 7.

In which part you did a verification of you proposed model ?

Reviewer 2 Report

In this work, the authors describe clear data processing methods to achieve food quality identification and prediction. Effectiveness of data analysis methods validated with public datasets. However, the author explained that the detection and processing of gas information is realized by using a smart E-nose. The article does not reflect the appearance of E-nose, the detection principle, etc. I hope the author can supplement it.

Reviewer 3 Report

Electronic noses (e-noses) are analytical technologies that mimic the biological noses, which are non-specific, low-selective, and cross-sensitive volatile organic compounds, VOCs, systems. These systems can be used for qualitative and quantitative fast-track detection of food adulteration. The subject is interesting and actual. The Authors have used sample slicing window protocol to extract the early part of the signals. This seems to me to be novelty in the subject.

 Detailed comments:

It seems to me that “Data Availability Statement: https://data.mendeley.com/datasets/5yhggs7zy7/1” is very crucial for the manuscript. It is mentioned at the end of the manuscript. It should be referenced just at the beginning of the experimental section.

It is not clear to me whether the Authors have performed their own experiments or they used the results from the literature. This should be clear stated.

If the Authors performed their own experiments, they should give more details about the experimental setup, i.e. the setup design, the selectivity of the sensors, how the humidity and temperature in the experimental setup was controlled, what computer programs were used to elaborate the measurment results and for statistics.

If they used the results of the other Authors, they should have the agreement to do so.

Reviewer 4 Report

Research on the issues related to the development of an artificial nose is absolutely justified and very necessary. They are in line with the recently dynamically developing direction of olfactometry. The creation of a durable and universal odour measurement transducer and the development of an effective method of analysing the results, and then classifying the smells, would be a very valuable tool for the development of many areas, for example in medical diagnostics, gastronomy or agriculture.

In general, the content of the article has been correctly divided into chapters according to the generally accepted method.

However, I have the impression that the scientific achievements described in this article are quite small. They only include the analysis of measurement data obtained from other studies. In fact, the article presents the results of the analysis obtained after interpreting the measurement signals with the use of various mathematical methods.

I also have also a few additional particular comments to the content of the presented article. The order of the comments does not reflect their significance. It results only from the order of appearance in the text of the article.

My remarks and comments:

1.       Lines 14-141, “MQ138, MQ137, MQ136, MQ135, MQ9, MQ6, MQ4, MQ2, and DHT22” - Details about these sensors are missing. Such information is crucial if we want to assess the accuracy of the proposed methods of odour recognition analysis. Information about the uncertainty of these sensors and their dynamic properties will allow a broader look at the values of statistical indicators presented later in the article and "SSWP-windows".

2.       Line 146, “the early part” - I believe that this term needs to be further explained in this section of the manuscript. Although reading the rest of the text you can guess what this term means, but nevertheless I think that at the very beginning of the text this term should be clarified.

3.       Line 150, “window” – the same as above.

4.       Line 164, “‘k’ is determined as 6” - Why 6 exactly? I expected 120=(60 samples / 0.5Hz).

5.       Line 181, “data normalization” - What were the min and max values for the 1st window? Were they the ones from the first window or the 6th? These values differ significantly for these windows.

6.       Line 191, “high” - Commonly referred to as "multi".

7.       Table 1, last row, left column, “Class-B” - It should probably be Class-C.

8.       Equation 7, ”1*20”  - Why is it not 2 * 10, etc. below.

9.       Line 270, “99.994% is obtained with entire data” - It needs to be commented on why the entire data is less accurate. The 99.99 level is very high anyway. Maybe we should not pay attention to the third decimal place and say that the results are the same. Perhaps the difference in the third decimal place is due to a random error in the measurement itself.

10.   Line 273, “thereby reducing recognition” - it would be very interesting to see the result for the remaining windows. I'm afraid the rest of the results are very close together.

11.   Line 279, “From the table” - a remark similar to the classification model SVM-C, it is very interesting what the differences between windows are.

12.   Line 286, “and the same is visible in Fig. 3.” - I suggest you give up the drawings; they present the same data already contained in the tables. Moreover, the charts are hardly legible. By the way - how the window number can be a non-integer, for example 1.5 or 2.5.

13.   Line 320, “the time required to train the two models is 0.51” - What is involved in the "train" process. The given time of 0.51s is a very short time for typical VOC sensors.

14.   Line 353, “to 95.20%” - Does this difference result from: the methodology of the results analysis, or from the metrological properties of the sensors used, or from the method of carrying out the measurement itself?

15.   Lines 362-363, “the colorimetric sensor array are exposed for five minutes to a food sample to detect VOCs from the food sample” - This requires an explanation of how the colorimetric sensor detected the VOC.

Reviewer 5 Report

The authors presented a design and developing of a smart electronic nose (SE-Nose) for qualitative and quantitative fast-track detection of food adulteration. Accuracy of 99.996%, RMSE of 0.02864 is 10 achieved with SVM classification and regression model. They reported that SE-Nose methodology, 11 the recognition time is reduced by one-third. To validate the classification and regression models, a 12 10-fold cross-validation method was also used. The manuscript is well organized. The presented paper is interesting, but the following corrections should be done before publishing.

Below are my concerns and suggestions to improve the manuscript,

The abstract needs to be highly quantitative.

Please provide more details for the SE-NOSE platform such as a figure or schematic design in the manuscript.

Please, explain the thermal stability and chemical stability of the platform. 

Reproducibility studies are very important for this device. The authors must explain the advantages. 

Please provide more details about the detection time and the sample volume. The authors must explain the advantages of the developed technique. 

The conclusion needs to be highly quantitative and should be discussed in more detail.

Round 2

Reviewer 5 Report

I think the author gave appropriate answers to my suggestions. Therefore, the article can be accepted in this form in the journal "Sensors"